# Disparities in distribution of COVID-19 vaccines across US counties: A geographic information system–based cross-sectional study

Inmaculada Hernandez[1]*, Sean Dickson[2], Shangbin Tang[1], Nico Gabriel[1], Lucas A. Berenbrok[3], Jingchuan Guo[4]

1 Division of Clinical Pharmacy, University of California, San Diego, Skaggs School of Pharmacy and Pharmaceutical Sciences, La Jolla, California, United States of America, 2 West Health Policy Center, Washington, DC, United States of America, 3 Department of Pharmacy and Therapeutics, University of Pittsburgh School of Pharmacy, Pittsburgh, Pennsylvania, United States of America, 4 Department of Pharmaceutical Outcomes and Policy, University of Florida College of Pharmacy, Gainesville, Florida, United States of America

* inhernandez@health.ucsd.edu

**Data Availability Statement:** Data from community pharmacies were obtained from the National Council for Prescription Drug Programs

## Abstract

### Background

The US Centers for Disease Control and Prevention has repeatedly called for Coronavirus Disease 2019 (COVID-19) vaccine equity. The objective our study was to measure equity in the early distribution of COVID-19 vaccines to healthcare facilities across the US. Specifically, we tested whether the likelihood of a healthcare facility administering COVID-19 vaccines in May 2021 differed by county-level racial composition and degree of urbanicity.

### Methods and findings

The outcome was whether an eligible vaccination facility actually administered COVID-19 vaccines as of May 2021, and was defined by spatially matching locations of eligible and actual COVID-19 vaccine administration locations. The outcome was regressed against county-level measures for racial/ethnic composition, urbanicity, income, social vulnerability index, COVID-19 mortality, 2020 election results, and availability of nontraditional vaccination locations using generalized estimating equations.

Across the US, 61.4% of eligible healthcare facilities and 76.0% of eligible pharmacies provided COVID-19 vaccinations as of May 2021. Facilities in counties with >42.2% non-Hispanic Black population (i.e., > 95th county percentile of Black race composition) were less likely to serve as COVID-19 vaccine administration locations compared to facilities in counties with <12.5% non-Hispanic Black population (i.e., lower than US average), with OR 0.83; 95% CI, 0.70 to 0.98, $p = 0.030$. Location of a facility in a rural county (OR 0.82; 95% CI, 0.75 to 0.90, $p < 0.001$, versus metropolitan county) or in a county in the top quintile of COVID-19 mortality (OR 0.83; 95% CI, 0.75 to 0.93, $p = 0.001$, versus bottom 4 quintiles)

under a license that does not allow for data sharing. This is the reason why the data cannot be shared publicly without restrictions. Requests for community pharmacy data should be addressed to the National Council for Prescription Drug Programs (http://dataq.ncpdp.org/) The remaining data sources are publically available: Addresses of federally qualified health centers are available from the Health Resources and Services Administration website: https://data.hrsa.gov/data/reports/datagrid?gridName=FQHCs Coordinates of rural health clinics are available from the Centers for Medicare and Medicaid Services website https://www.cms.gov/Research-Statistics-Data-and-Systems/Downloadable-Public-Use-Files/Cost-Reports/Rural-Health-Center-222-2017-form Addresses of hospital outpatient departments are also available from the Centers for Medicare and Medicaid Services website https://www.cms.gov/Research-Statistics-Data-and-Systems/Statistics-Trends-and-Reports/Medicare-Provider-Charge-Data/Downloads/Outpatient_Data_2017_XLSX.zip Addresses of COVID-19 vaccine administration locations are available from the Centers for Disease Control and Prevention website https://www.cdc.gov/vaccines/covid-19/reporting/vaccinefinder/about.html.

**Funding:** The author(s) received no specific funding for this work.

**Competing interests:** I have read the journal's policy and the authors of this manuscript have the following competing interests: IH reports consulting fees from Pfizer, outside of the submitted work.

**Abbreviations:** CDC, Centers for Disease Control and Prevention; COVID-19, Coronavirus Disease 2019; FQHC, federally qualified health center; HOD, hospital outpatient department; RHC, rural health clinic; RUCC, Rural–Urban Continuum Codes; SARS-CoV-2, Severe Acute Respiratory Syndrome Coronavirus 2.

was associated with decreased odds of serving as a COVID-19 vaccine administration location.

There was a significant interaction of urbanicity and racial/ethnic composition: In metropolitan counties, facilities in counties with >42.2% non-Hispanic Black population (i.e., >95th county percentile of Black race composition) had 32% (95% CI 14% to 47%, $p = 0.001$) lower odds of serving as COVID administration facility compared to facilities in counties with below US average Black population. This association between Black composition and odds of a facility serving as vaccine administration facility was not observed in rural or suburban counties. In rural counties, facilities in counties with above US average Hispanic population had 26% (95% CI 11% to 38%, $p = 0.002$) lower odds of serving as vaccine administration facility compared to facilities in counties with below US average Hispanic population. This association between Hispanic ethnicity and odds of a facility serving as vaccine administration facility was not observed in metropolitan or suburban counties.

Our analyses did not include nontraditional vaccination sites and are based on data as of May 2021, thus they represent the early distribution of COVID-19 vaccines. Our results based on this cross-sectional analysis may not be generalizable to later phases of the COVID-19 vaccine distribution process.

## Conclusions

Healthcare facilities in counties with higher Black composition, in rural areas, and in hardest-hit communities were less likely to serve as COVID-19 vaccine administration locations in May 2021. The lower uptake of COVID-19 vaccinations among minority populations and rural areas has been attributed to vaccine hesitancy; however, decreased access to vaccination sites may be an additional overlooked barrier.

## Author summary

### Why was this study done?

- Equity in the distribution of Coronavirus Disease 2019 (COVID-19) vaccine is of major relevance.
- It is unknown whether there were differences in the distribution of COVID-19 vaccines to healthcare facilities depending on the demographic composition of the population.

### What did the researchers do and find?

- We tested whether healthcare facilities serving minority or disadvantaged neighborhoods were less likely to administer COVID-19 vaccines in the early phase of the COVID-19 vaccine rollout process.
- We found that healthcare facilities in counties with higher Black composition, in rural areas, and in hardest-hit communities were less likely to administer COVID-19 vaccines in May 2021.

**What do these findings mean?**

- There were disparities in the early distribution of COVID-19 vaccines to healthcare facilities across the country.

## Introduction

The equitable and timely distribution of Coronavirus Disease 2019 (COVID-19) vaccines was the public health priority in 2021, after the successful race to develop vaccines against Severe Acute Respiratory Syndrome Coronavirus 2 (SARS-CoV-2) in 2020. In the United States, the distribution of COVID-19 vaccines followed the framework proposed by the National Academies of Science, Engineering, and Medicine, which prioritized in early phases of vaccine distribution healthcare workers and long-term care residents, followed by older adults, frontline workers, those with high-risk conditions, and essential workers [1]. For each population group, the framework called for a prioritization of vulnerable areas that have been disproportionately impacted by the COVID-19 pandemic. Although the US public generally agreed with the prioritization of disadvantaged communities hardest hit by COVID in vaccination [2], most COVID-19 distribution plans were created without input from health equity experts [3].

Consistently across the phases of distribution of COVID-19 vaccines, vaccine uptake has been lower among racial/ethnic minority groups than non-Hispanic White individuals [4–6]. Mistrust, misinformation, and access to COVID-19 vaccines have been identified as the key determinants of uptake of COVID-19 vaccinations among underrepresented groups [7]. However, most discussions on vaccine uptake among minority populations have focused on mistrust and misinformation [7–11]. Soon after the approval of COVID-19 vaccines, we reported that spatial access to healthcare should be a major consideration in ensuring an equitable access to COVID-19 vaccines because underrepresented minorities are less likely to live near healthcare facilities than non-Hispanic White individuals [12]. Rader and colleagues recently reported that COVID-19 vaccine deserts were more likely to be located in rural and lower-income areas [13]. Similar findings on the distribution of vaccine sites were noted by Williams and colleagues for the borough of Brooklyn, New York [14]. In the state of Florida, Kim and colleagues also observed lower accessibility of underserved communities to COVID-19 vaccination sites [15].

These past studies adopted a population focus in comparing the accessibility to COVID-19 vaccination sites across sociodemographic subgroups. Such approach is, however, limited in that it does not differentiate whether lower access in underserved neighborhoods is a product of the lower concentration of healthcare facilities in these areas or of inequities in the distribution of COVID-19 vaccines to facilities. To answer this question, we tested whether the likelihood of an eligible healthcare facility administering COVID-19 vaccines in May 2021 varied with the county-level racial/ethnic composition and degree of urbanicity. This approach enabled us to quantify equity in the early distribution of COVID-19 vaccines to facilities. We hypothesized that healthcare facilities serving rural areas and underrepresented populations would be less likely to administer COVID-19 vaccines.

## Methods

### Data sources

In a prior analysis, we identified and mapped open-door healthcare facilities eligible for the delivery of COVID-19 vaccinations, including community pharmacies licensed to provide

immunizations, federally qualified health centers (FQHCs), rural health clinics (RHCs), and hospital outpatient departments (HODs) [12]. Long-term facilities were not included because they are not available for the vaccination of community dwelling individuals. Nontraditional settings for vaccination such as stadiums or convention centers were not included since they are not healthcare facilities.

Data from community pharmacies were obtained from the National Council for Prescription Drug Programs. Addresses of FQHCs were obtained from the Health Resources and Services Administration [16]. Coordinates of RHCs and addresses of HODs were obtained from the Centers for Medicare and Medicaid Services [17,18]. To identify actual COVID-19 vaccine administration locations, we extracted VaccineFinder data from the Centers for Disease Control and Prevention (CDC) as of May 10, 2021 [19]. These data capture the healthcare facilities involved in the early distribution of COVID-19 vaccines, since the federal government required states to make vaccines available to the general public by May 1, 2021 [20].

Data from community pharmacies were obtained by a restricted license from the National Council for Prescription Drug Programs, but the remaining data sources are publicly available as detailed in the Data Statement. Our study was exempt from human subjects regulation because no human data were used. The study did not have a registered protocol, but it is reported as per the Strengthening the Reporting of Observational Studies in Epidemiology (STROBE) guideline (S1 STROBE Checklist). Analyses were performed June to September 2021.

## Outcome

The outcome was whether an eligible vaccination facility was registered as actually administering COVID-19 vaccines to the public in May 2021, and was defined by spatially matching locations of eligible and actual COVID-19 vaccine administration locations. Two locations were considered a match if (1) they had the same placekey (a universal standard identifier for any physical location); or (2) they were closest in proximity to each other after geocoding, and their addresses and names matched. Geocoding was performed using ArcGIS, version 10.7 (Esri).

## Independent variables

Independent variables included racial/ethnic composition, urbanicity, median household income, social vulnerability index, COVID-19 mortality rate, 2020 presidential election results, and availability of other COVID-19 vaccine administration locations. Independent variables were defined at the county level as opposed to the census tract level because first, counties are the political unit with most control over local vaccine distribution, and second, service areas of HODs, FQHCs, and RHCs span multiple census tracts.

County-level measures of racial/ethnic composition included proportion non-Hispanic Black and proportion Hispanic and were obtained from the University of Wisconsin Population Health Institute County Rankings and Roadmap data [21]. Counties were categorized in 3 levels based on racial/ethnic composition: (1) below US average; (2) between US average and 95th county percentile; and (3) above 95th county percentile, as previously done by CDC [22].

Urbanicity was categorized in 3 levels using US Department of Agriculture Rural–Urban Continuum Codes (RUCC): (1) metropolitan (RUCC code 1, accounts for 168.5 million people); (2) suburban (RUCC codes 2 to 3, accounts for 93.9 million people); and (3) rural (RUCC codes 4 to 9, accounts for 46.3 million people) [23].

Social vulnerability index was obtained from the CDC and represents the degree to which a community exhibits certain conditions that affect its ability to prevent human suffering and

financial loss in the event of a disaster [24]. The social vulnerability index includes 4 components: socioeconomic, household composition and disability, minority status and language, and housing type and transportation. We created indicator variables for counties in the top quintile of the socioeconomic, household composition and disability, and housing type and transportation components. These indicator variables represent most vulnerable communities. We did not include a variable for the minority status and language component of the social vulnerability index because minority status is already captured by the measures of racial/ethnic composition defined above.

COVID-19 mortality was obtained as of May 10, 2021, from the John Hopkins University Coronavirus Resource Center [25]. We created an indicator variable for counties in the top quintile of COVID-19 mortality, which represent communities that were hardest-hit by COVID-19 prior to the date of our analysis.

Results of the 2020 election were obtained from the MIT Election Data and Science Lab [26]. We created an indicator variable representing the Presidential candidate that won the 2020 election in each county. This variable was included in analysis to capture the majority political affiliation of the area served by a facility.

Healthcare facilities in counties relying on nontraditional settings for vaccination such as stadiums or convention centers may have been less likely to administer vaccines. To account for this, we adjusted analyses for the density of alternative vaccination locations. Alternative vaccine administration locations were defined as vaccination locations reported by CDC in VaccineFinder data that did not match to pharmacies, HODs, RHCs, or FQHCs. Finally, a healthcare facility may have been less likely to administer COVID-19 vaccines if there was another COVID-19 vaccination location nearby. Thus, we adjusted for the availability of a COVID vaccine administration facility within 500 m of a given facility.

## Statistical analyses

The outcome—whether an eligible vaccination facility was registered as actually administering COVID-19 vaccines in May 2021—was regressed against covariates listed above using logistic regression. All independent variables specified above were included in the final model. We applied generalized estimating equations to account for clustering of facilities within a county. We tested interactions between racial/ethnic composition and urbanicity. We constructed 1 set of analyses for all 4 types of healthcare facilities and a second set for pharmacies because some FQHCs, RHCs, and HODs only offered COVID-19 vaccines to registered patients. Analyses were conducted using SAS 9.4 (Cary, North Carolina) and Stata 17 (College Station, Texas).

## Results

The sample included 50,806 community pharmacies, 11,619 FQHCs, 3,187 HOPDs, and 1,255 RHCs distributed across 2,942 counties in US states (territories were not included). Across the US, 61.4% of eligible healthcare facilities and 76.0% of eligible pharmacies provided COVID-19 vaccinations in May 2021 (**Table 1**).

In the early phase of the vaccine rollout process, healthcare facilities were less likely to serve as COVID-19 vaccine administration locations when located in counties with >42.2% Black population (i.e., > 95th county percentile of non-Hispanic Black race composition) compared to counties with <12.5% non-Hispanic Black population (i.e., lower than US average), with OR 0.83; 95% CI, 0.70 to 0.98, $p = 0.030$ (**Fig 1**). Location of a facility in a rural county (OR 0.82; 95% CI, 0.75 to 0.90, $p < 0.001$, versus metropolitan county) or in a county in the top quintile of COVID-19 mortality (OR 0.83; 95% CI, 0.75 to 0.93, $p = 0.001$, versus bottom 4

**Table 1. Proportion of facilities serving as COVID-19 vaccine administration locations.**

| Variable—n(%) | % Served as COVID-19 Vaccine Administration Location | |
| --- | --- | --- |
| | Pharmacies ($n$ = 50,806) | All Healthcare Facilities[a] ($n$ = 66,867) |
| All | 76.0% | 61.4% |
| County-Level Proportion of Black Population[b] | | |
| Facility in County with Proportion Black Population <12.5% | 76.3% | 60.5% |
| Facility in County with Proportion Black Population 12.5%–42.2% | 75.9% | 63.9% |
| Facility in County with Proportion Black Population >42.2% | 71.8% | 55.5% |
| County-Level Proportion of Hispanic Population[c] | | |
| Facility in County with Proportion Hispanic Population <18.5% | 75.8% | 62.2% |
| Facility in County with Proportion Hispanic Population 18.5%–38.7% | 76.4% | 61.5% |
| Facility in County with Proportion Hispanic Population >38.7% | 76.4% | 56.4% |
| Urbanicity[d] | | |
| Metropolitan | 76.4% | 64.4% |
| Suburban | 77.4% | 62.7% |
| Rural | 71.7% | 51.1% |
| Facility in County in Bottom Quintile for Median Income | 71.8% | 45.5% |
| Facility in County in Top Quintile for Vulnerability Index—Socioeconomic Component | 73.5% | 48.8% |
| Facility in County in Top Quintile for Vulnerability Index—Household Composition and Disability | 72.6% | 51.2% |
| Facility in County in Top Quintile for Vulnerability Index -Housing Type and Transportation | 74.5% | 55.8% |
| Facility in County in Top Quintile for COVID Mortality | 71.4% | 54.6% |
| Facility in County where Trump Won the 2020 presidential election | 75.3% | 60.0% |

[a]Included pharmacies, FQHCs, RHCs, and HODs.

[b]The proportion of Black population at the county level was obtained from the University of Wisconsin Population Health Institute County Rankings and Roadmap data[21] and was categorized in 3 levels, following prior methodology used by CDC [22]: (1) below US average (12.5%); (2) between US average and 95th county percentile (42.2%); and (3) above 95th county percentile (42.2%).

[c]The proportion of Hispanic population at the county level was obtained from the University of Wisconsin Population Health Institute County Rankings and Roadmap data [21] and was categorized in 3 levels, following prior methodology used by CDC [22]: (1) below US average (18.5%); (2) between US average and 95th county percentile (38.7%); and (3) above 95th county percentile (38.4%).

[d]Urbanicity was categorized in 3 levels using US Department of Agriculture RUCC: (1) metropolitan (RUCC code 1); (2) suburban (RUCC codes 2–3); and (3) rural (RUCC codes 4–9).

CDC, Centers for Disease Control and Prevention; COVID-19, Coronavirus Disease 2019; FQHC, federally qualified health center; HOD, hospital outpatient department; RHC, rural health clinic; RUCC, Rural–Urban Continuum Codes.

quintiles) was associated with decreased odds of serving as a COVID-19 vaccine administration location. There were no differences in the likelihood of facilities serving as COVID-19 vaccine administration locations by voting results in the 2020 Presidential election (OR 0.96; 95% CI, 0.89 to 1.03, $p$ = 0.240 for counties with majority vote for Trump, compared to

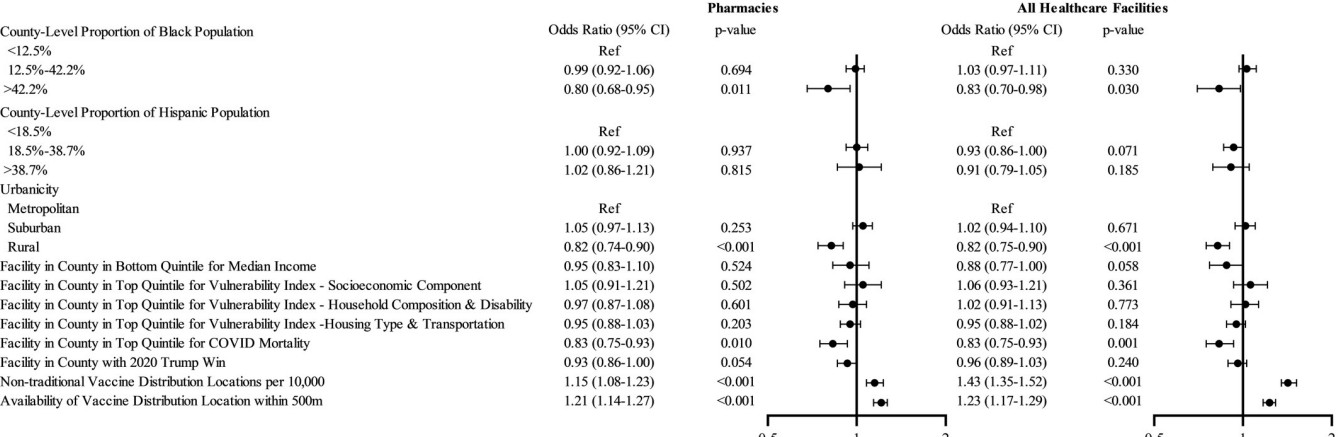

**Fig 1. Adjusted odds ratios of facilities serving as COVID-19 vaccine administration locations, main effects.** The figure shows the results of logistic regression models fitted with generalized estimating equations for the primary outcome of a healthcare facility (or a pharmacy) serving as a COVID-19 vaccine administration location. The model only included main effects. All healthcare facilities included pharmacies, FQHCs, RHCs, and HODs. The circles represent the point estimate for the odds ratio, and the whiskers represent the 95% confidence interval. COVID-19, Coronavirus Disease 2019; FQHC, federally qualified health center; HOD, hospital outpatient department; RHC, rural health clinic.

counties with majority vote for Biden). Results were similar for analyses conducted for community pharmacies.

We identified a significant interaction of urbanicity with non-Hispanic Black race composition: In metropolitan counties, facilities in counties with >42.2% non-Hispanic Black population (i.e., > 95th county percentile of Black race composition) had 32% (95% CI 14% to 47%, $p = 0.001$) lower odds of serving as COVID administration facility compared to facilities in counties with below US average Black population (Fig 2). This disparity was not observed in rural or suburban counties. Moreover, we detected a gradient for pharmacies: Pharmacies in metropolitan counties with above-average non-Hispanic Black composition had 12% (95% CI, 1% to 20%, $p = 0.030$) lower odds of serving as COVID-19 vaccine administration locations than pharmacies in metropolitan counties with below-average Black composition; in counties above the top 95th percentile of non-Hispanic Black composition, odds were 30% (95% CI, 11% to 41%, $p = 0.003$) lower.

We also identified a significant interaction of urbanicity with Hispanic composition: In rural counties with above-average Hispanic composition, healthcare facilities had 26% (95% CI, 11% to 38%, $p = 0.002$) lower odds of serving as COVID-19 vaccine administration locations compared to facilities in counties with below-average Hispanic composition (Fig 3). However, this disparity was not observed in metropolitan or suburban counties.

## Discussion

To our knowledge, we present the first nationwide study to quantify disparities in the early distribution of COVID-19 vaccines to healthcare facilities in the US. Our study demonstrates that healthcare facilities in counties with higher non-Hispanic Black composition, in rural areas, and in hardest-hit communities were less likely to serve as COVID-19 vaccine administration locations in May 2021. We observed, however, significant interactions between urbanicity and demographic composition: The county-level proportion of Black population was associated

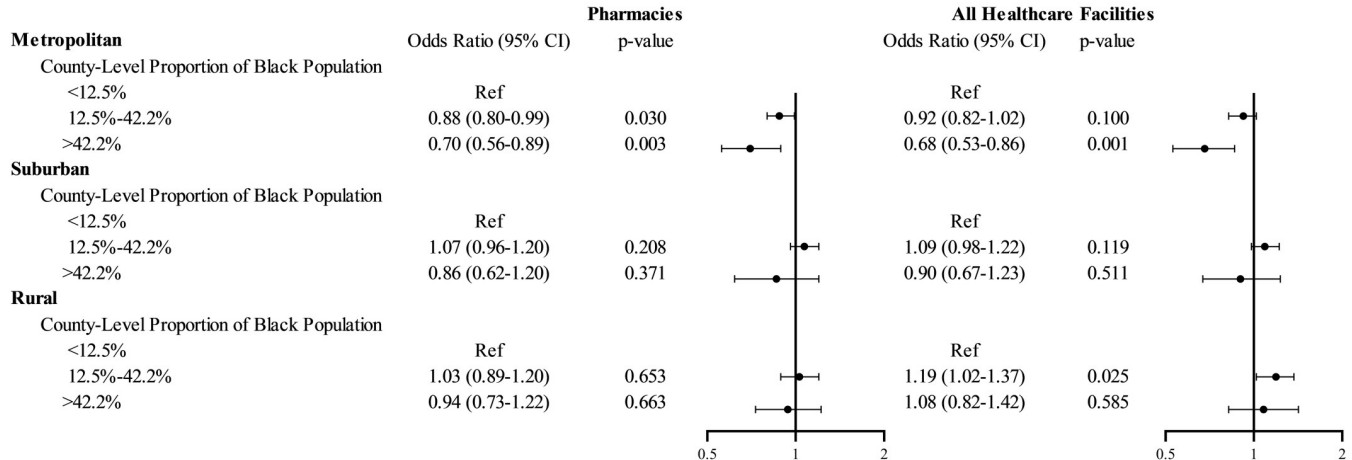

**Fig 2. Adjusted odds ratios of facilities serving as COVID-19 vaccine administration locations, interaction for proportion non-Hispanic Black population and urbanicity.** The figure shows the results of logistic regression models fitted with generalized estimating equations for the primary outcome of a healthcare facility (or a pharmacy) serving as a COVID-19 vaccine administration location. All healthcare facilities included pharmacies, FQHCs, RHCs, and HODs. The model adjusted for all covariates listed in Fig 1. Additionally, the model constructed for all healthcare facilities included an indicator variable for facility type (pharmacy vs. others). The circles represent the point estimate for the odds ratio, and the whiskers represent the 95% confidence interval. COVID-19, Coronavirus Disease 2019; FQHC, federally qualified health center; HOD, hospital outpatient department; RHC, rural health clinic.

with decreased odds of a facility administering COVID-19 vaccines in metropolitan, but not in suburban or rural counties. The county-level proportion of Hispanic population was associated with decreased odds of a facility administering COVID-19 vaccines in rural, but not in

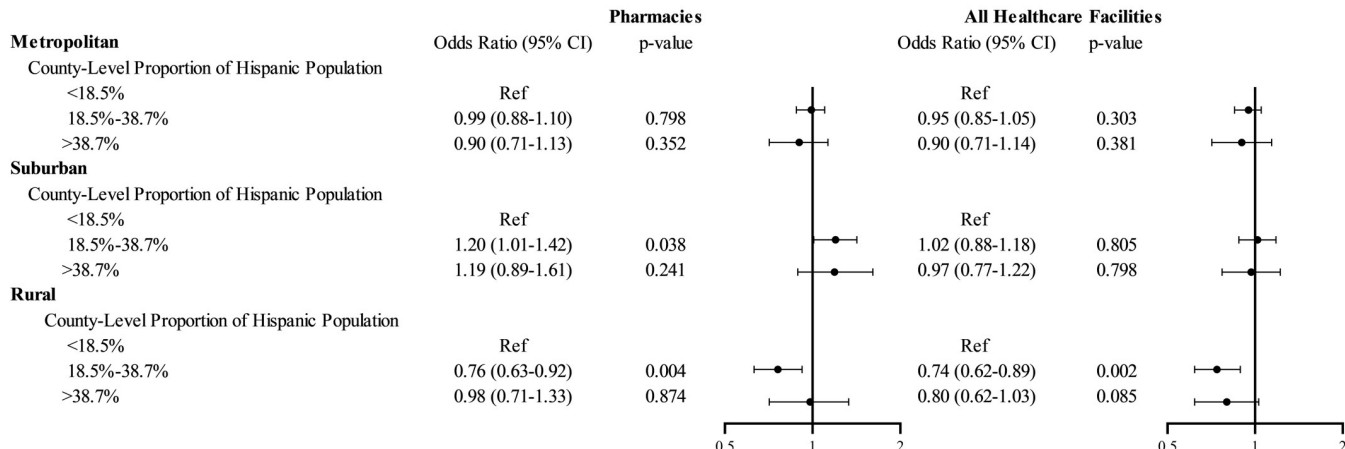

**Fig 3. Adjusted odds ratios of facilities serving as COVID-19 vaccine administration locations, interaction for proportion Hispanic population and urbanicity.** The figure shows the results of logistic regression models fitted with generalized estimating equations for the primary outcome of a healthcare facility (or a pharmacy) serving as a COVID-19 vaccine administration location. All healthcare facilities included pharmacies, FQHCs, RHCs, and HODs. The model adjusted for all covariates listed in Fig 1. Additionally, the model constructed for all healthcare facilities included an indicator variable for facility type (pharmacy vs. others). The circles represent the point estimate for the odds ratio, and the whiskers represent the 95% confidence interval. COVID-19, Coronavirus Disease 2019; FQHC, federally qualified health center; HOD, hospital outpatient department; RHC, rural health clinic.

metropolitan or suburban counties. Our findings based on May 2021 data represent the early distribution of COVID-19 vaccines and may not be generalizable to later phases of the COVID-19 vaccine rollout process.

A study by Kim and colleagues examined disparities in access to COVID-19 facilities in Florida and found that racial/ethnic disparities in access differed across rural and urban counties [15]. Consistent with our findings, this study reported that, compared to non-Hispanic White individuals, Black individuals had lower access to COVID-19 vaccine administration sites in urban counties; however, disparities in access for the Hispanic population were concentrated in rural counties [15]. Our findings are also consistent with the results by Rader and colleagues, who found that COVID-19 vaccine deserts are more likely to be located in rural and low-income areas [13]. Nevertheless, our analysis is a major contribution to the existing literature because, to our knowledge, it is the first to analyze equity in the distribution of vaccines to healthcare facilities across the nation. This is important because when studies evaluate population access to COVID-19 administration facilities, it is not possible to differentiate whether lower access in rural or underserved areas is a product of the lower concentration of healthcare facilities or of inequities in the distribution of COVID-19 vaccines to facilities. To answer this question, we tested instead whether the likelihood of healthcare facilities serving as COVID-19 administration sites in May 2021 differed with the sociodemographic composition of the population served. In doing so, we demonstrate that facilities in counties with higher non-Hispanic Black composition, in rural areas, and in hardest-hit communities were less likely to serve as COVID-19 vaccine administration locations in the early phase of the vaccine rollout process. In other words, we demonstrate that underrepresented, underserved, and rural areas have lower access to COVID-19 vaccines not only because of the lower concentration of healthcare facilities, but also because the facilities serving these areas were less likely to administer COVID-19 vaccines, at least in the early phase of the vaccine distribution process.

Our analyses were conducted at the facility level, but covariates were defined at the county level for 2 reasons: First, counties are the political unit with most control over local vaccine distribution; second, facility service areas can span hundreds of census tracts [27]. Our sample did not include nontraditional vaccination sites because our objective was to measure the likelihood of existing healthcare facilities serving as COVID-19 vaccine administration locations. Nevertheless, we adjusted for the density of alternative vaccination locations. In other words, we accounted for the fact that healthcare facilities in counties heavily relying on nontraditional settings may have been less likely to administer COVID-19 vaccines. Our evaluation of racial/ethnic disparities focused on non-Hispanic Black and Hispanic individuals due to the low proportion of other minority groups in most counties, which yielded unstable estimates.

An equitable distribution of vaccines would imply that vaccines were distributed in a timely manner across all individuals. This is the reason why we used data from May 2021, when every US adult became eligible for vaccination. The association between vaccine distribution and sociodemographics of the population served by facilities could have changed over time. In other words, our findings based on May 2021 data may not be generalizable to later phases of the COVID-19 vaccine rollout process. Nevertheless, our findings based on the initial rollout process are of prominent relevance, because analyses based on more recent data could mask differences in the timing of vaccine distribution across facilities.

Our nationwide quantitative evaluation of health equity in the early distribution of COVID-19 vaccines is limited by the data available. We operationalized the definition of health inequity as differences in the likelihood of a healthcare facility administering vaccines associated with the sociodemographics of the population served. This binary outcome does not account for important variables not available in the data such as the number of vaccines distributed in each healthcare facility, the volume of vaccines distributed in other facilities

within the county, or the area of service of each facility. One could argue that the likelihood of a facility administering vaccines is not relevant because an area could be well covered by a smaller fraction of facilities, if such facilities provided enough vaccines to cover the population and were accessible enough. Nevertheless, regardless of whether a smaller share of facilities is able to adequately cover the population through higher volume, the variation in the proportion of available healthcare facilities used associated with the sociodemographic characteristics of the population suggests structural inequities in the design of the early COVID-19 vaccine roll-out. The differential use of available healthcare infrastructure by the demographics of the population served may suggest prioritization of convenience for some residents rather than the actual distribution volume needed.

Four additional limitations of our analyses are worth noting. First, locations not registered in VaccineFinder were not identified, which could have led to an underestimation of the proportion of healthcare facilities serving as vaccine administration locations. Second, we did not evaluate variation in the types of vaccines distributed to each healthcare facility. Third, the data compiled only contained information on the addresses of vaccine administration sites, so it was impossible to define whether institutional or structural reasons led some facilities to not become COVID-19 administration sites. Finally, our data and analyses were at the facility level rather than at the person level, so we were not able to assess individual preferences in accessing healthcare facilities for vaccination. Our study, however, presents important strengths, including the use of spatial matching methods and the national coverage of the results. The innovative approach used to measure health equity should also be noted, since we used facilities as observations, instead of individuals or census tracts.

The discussion on the lower uptake of COVID-19 vaccines among racial/ethnic minority groups has mostly focused on mistrust and misinformation [7–11]. However, our analysis suggests that systematic barriers play an important role in differential rates of COVID-19 vaccination across racial/ethnic groups, which are often omitted in conversations around mistrust. In addition to the lower concentration of healthcare facilities in underserved and rural areas [12], the facilities that serve these vulnerable populations were less likely to administer COVID-19 vaccines in the early phase of the vaccine rollout process, even when the framework proposed by the National Academies of Science, Engineering, and Medicine called for a prioritization of these areas precisely [1]. In tandem with community engagement efforts to address vaccine hesitancy and interventions to improve spatial access to vaccines, public health authorities should review COVID-19 distribution plans to identify the reasons why these processes resulted in an inequitable distribution of COVID-19 vaccines in early 2021. Identifying the drivers of the lower involvement of facilities serving underserved and rural areas in the early distribution of the COVID-19 vaccine is crucial to improve equity in the distribution of booster shots and of future public health prevention programs.

We evaluated whether the likelihood of healthcare facilities serving as COVID-19 vaccine administration sites in the early distribution of COVID-19 vaccines varied with the sociodemographic composition of the population served. We found that healthcare facilities in counties with higher Black composition, in rural areas, and in hardest-hit communities were less likely to serve as COVID-19 vaccine administration locations in the early phase of the vaccine rollout process.

## Supporting information

**S1 STROBE Checklist. STROBE checklist for cross-sectional study.**
(DOC)

## Acknowledgments

We thank the Data Science team at the West Health Institute for their help extracting Vaccine-Finder data.

## Author Contributions

**Conceptualization:** Inmaculada Hernandez, Sean Dickson, Lucas A. Berenbrok, Jingchuan Guo.

**Data curation:** Shangbin Tang, Nico Gabriel.

**Formal analysis:** Shangbin Tang, Nico Gabriel.

**Funding acquisition:** Inmaculada Hernandez.

**Investigation:** Inmaculada Hernandez, Lucas A. Berenbrok, Jingchuan Guo.

**Methodology:** Sean Dickson, Shangbin Tang, Nico Gabriel, Jingchuan Guo.

**Project administration:** Inmaculada Hernandez, Jingchuan Guo.

**Resources:** Inmaculada Hernandez, Lucas A. Berenbrok.

**Software:** Nico Gabriel.

**Supervision:** Inmaculada Hernandez, Sean Dickson, Jingchuan Guo.

**Visualization:** Nico Gabriel.

**Writing – original draft:** Inmaculada Hernandez.

**Writing – review & editing:** Sean Dickson, Shangbin Tang, Nico Gabriel, Lucas A. Berenbrok, Jingchuan Guo.

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
