## [Editor Report · Decision Letter 0]

27 Oct 2021

Dear Dr Hernandez, 

Thank you for submitting your manuscript entitled "Disparities in Distribution of COVID-19 Vaccines across U.S. Counties" for consideration by PLOS Medicine.

Your manuscript has now been evaluated by the PLOS Medicine editorial staff and I am writing to let you know that we would like to send your submission out for external peer review.

Please re-submit your manuscript within two working days, i.e. by Oct 29 2021 11:59PM.

Kind regards,

Beryne Odeny

PLOS Medicine

---

## [Decision Letter · Decision Letter 1]

17 Mar 2022

Dear Dr. Hernandez,

Thank you very much for submitting your manuscript "Disparities in Distribution of COVID-19 Vaccines across U.S. Counties" (PMEDICINE-D-21-04494R1) for consideration at PLOS Medicine. 

[LINK]

In light of these reviews, I am afraid that we will not be able to accept the manuscript for publication in the journal in its current form, but we would like to consider a revised version that addresses the reviewers' and editors' comments. Obviously we cannot make any decision about publication until we have seen the revised manuscript and your response, and we plan to seek re-review by one or more of the reviewers. 

We expect to receive your revised manuscript by Apr 07 2022 11:59PM. Please email us (plosmedicine@plos.org) if you have any questions or concerns.

We look forward to receiving your revised manuscript. 

Sincerely,

Beryne Odeny, 

PLOS Medicine

plosmedicine.org

1) Please revise your title according to PLOS Medicine's style. Your title must be nondeclarative and not a question. It should begin with main concept if possible. Please place the study design in the subtitle (i.e., after a colon), e.g., a retrospective cohort study.

2) Please include line numbers in your next draft.

3) Is there a chance you can obtain more recent data from this setting?

4) Abstract:

a) Please ensure that all numbers presented in the abstract are present and identical to numbers presented in the main manuscript text.

b) Please quantify the main results (please present both 95% CIs and p values).

c) In the last sentence of the Abstract Methods and Findings section, please describe the main limitation(s) of the study's methodology.

6) Introduction: Please address and cite past research and explain the need for and potential importance of your study. Indicate whether your study is novel and how you determined that. If there has been a systematic review of the evidence related to your study (or you have conducted one), please refer to and reference that review and indicate whether it supports the need for your study. 

7) Please conclude the “Introduction” with a clear description of the study question or hypothesis. The description of the database can be moved to the methods section

8) Did your study have a prospective protocol or analysis plan? Please state this (either way) early in the Methods section. 

9) Please temper claims of primacy of results (e.g. “In the first nation-wide study…)” by stating, "to our knowledge" or something similar.

10) Please ensure that the study is reported according to the STROBE and include the completed STROBE checklist as Supporting Information. Please add the following statement, or similar, to the Methods: "This study is reported as per the Strengthening the Reporting of Observational Studies in Epidemiology (STROBE) guideline (S1 Checklist)."

11) How was race/ethnicity defined and by whom?

12) You examined (as a binary outcome, yes/no) whether a facility serves as a covid-19 vaccination site; is there any data on the number of vaccines actually delivered?

13) Do you have any insights into the reasons why particular sites might be signed up as vaccination sites or not

14) Please discuss your analyses based on a hypothetical “Trump win” in 2020. For example, add a sentence to the effect that there were “no differences by voting preferences in the 2020 Presidential election”, or similar, assuming this is the case.

15) Please provide p values in addition to 95% CIs in the main text and tables

16) Please indicate in the figure caption the meaning of the bars and whiskers in Figure 1.

17) Please present and organize the Discussion as follows: a short, clear summary of the article's findings; what the study adds to existing research and where and why the results may differ from previous research; include relevant citations; strengths and limitations of the study; implications and next steps for research, clinical practice, and/or public policy; one-paragraph conclusion.

18) References: 

a) Please provide access dates for all references with a weblink

b) Please select the PLOS Medicine reference style in your citation manager. In-text reference call outs should be presented as follows noting the absence of spaces within the square brackets, 

c) Please ensure that journal name abbreviations consistently match those found in the National Center for Biotechnology Information (NCBI) databases. https://journals.plos.org/plosmedicine/s/submission-guidelines#loc-references. 

Notes from the Academic Editor:

The message and finding are important – that access is an issue. 

The major limitation is the ecological analysis, and I would think important to ensure you have addressed things like Social Vulnerability Index (SVI). Adjustment for SVI may attenuate the race-access relationship, but I think it would not necessarily undermine the argument – it could be a mediator of the relationship with racialization.

Comments from the reviewers:

Reviewer #1: The authors have applied geographic information systems methods to test whether the likelihood of an eligible healthcare facility administering COVID-19 vaccines varied with the county-level racial composition and degree of urbanicity.

Comments:

"The outcome was regressed against county-level measures for racial/ethnic composition, urbanicity, income, COVID-19 mortality, 2020 election results, and availability of non-traditional vaccination locations using generalized estimating equations."

Did the authors consider including covariates in the models in order to adjust for potential confounding, such as county- or facility- level age distribution, comorbidity prevalence, and gender ratio?

"The outcome -whether an eligible vaccination facility was registered as actually administering COVID19 vaccines—was regressed against covariates listed above using logistic regression. We applied generalized estimating equations to account for clustering of facilities within a county."

The authors have applied a technically appropriate modelling technique for the data and research question in hand.

"We tested interactions between racial/ethnic composition and urbanicity. Because some FQHCs, RHCs, and HODs only offered COVID-19 vaccines to registered patients, we constructed one set of analyses for all four types of healthcare facilities and a second set for pharmacies only."

The authors have undertaken several additional analyses that help to demonstrate the robustness of the study findings. 

Can the authors please present statistical findings (i.e. exact p-values and CIs) for all stated study inferences throughout the Results text?

Did the authors consider exploring 'County-Level Proportion of Black Population' and 'County with Proportion Hispanic Population' as continuous variables within the analysis?

Reviewer #2: I enjoyed reading this manuscript and I believe it is worthy of acceptance and publication in PLOS Medicine. I liked how the authors side-stepped the limitations regarding what data the federal government has regarding reported vaccination administrations with regards to race and ethnicity. However, before publication, I believe the manuscript can be strengthened. Most, if not all, of these suggestions should not require reanalysis, but may require some reorganization and a few more sentences. I think there is enough to call it a major revision, but hopefully not one that is a heavy lift:

The authors state their goal is to measure health equity in the "actual distribution of COVID-19 vaccines." If that is the ultimate goal, then the portion that is pharmacies or alternate facilities is less important than the overall totality of vaccination sites and how those impact minorities. If the goal is also to highlight that, after COVID-19 ends, there may be pullback and these variations in how vaccines are administered will be more important for 'normal' vaccinations, this could be stated more clearly. I suggest this is a secondary focus because of a sentence in the discussion that states the objective was to measure the likelihood of existing healthcare facilities providing vaccinations. This objective is different from the "actual distribution of vaccines" and makes the later discussion of improved geographic access to vaccination fall somewhat flat. For example, yes, there is a continuum for pharmacies for the Black population (Figure 1a), but for whatever reason this only holds significantly for the greater than 42.7% group for all health care facilities (and here that confidence interval is a 'squeaker.') The same issue for pharmacies versus all four types of facilities does not seem to exist for either the Rural or the COVID mortality. Pragmatically, if the only goal is to get COVID vaccines into arms, then who is delivering those vaccines does not matter. If you want to draw conclusions about what it looked like at the beginning of the pandemic, what the current coverage says about how the system is responding to inequities (and here you'd need to spell out why a pharmacy vaccination has a different connotation than a federal facility, etc.), or how vaccinations will look after, then who is giving the vaccines matters more.

Continuing on this theme, the authors do acknowledge non-traditional vaccine distribution sites and availability of other vaccine distribution locations, but as a reader I don't think I fully understood the implications in the text. Here, I'm trying to highlight reader confusion, rather than give exact instructions on how to address that confusion. What do these variable these mean along the rural to urban spectrum. What does 500m mean in a rural location? Is there some kind of relationship for rural locations having non-traditional sites? This is not primary to the manuscript, but I didn't feel comfortable that I understood what these variables really meant. They appear again in Figure 1A. Since I don't know what they mean, when I read their odds ratios, I am guessing that having them in a county means more people are getting vaccinated than one would expect? Does this imply they are likely geographic overcompensation? Does the philosophy of vaccine distribution sites differ for low density rural areas, where one would expect long drives to pretty much everywhere? Or does it not differ because you can't expect everyone to have cars?

Another issue that should be addressed, if not accounted for possibly, is that CDC and Census report "Black" as a race and "Hispanic" as an ethnicity. This means that Hispanic includes Hispanic Blacks and Black includes Hispanic Blacks. I am guessing that the authors ignored this nuance because this group of people is not large compared to the total for either Black or Hispanic. Or because, for example, Florida doesn't report its data like Census, etc. This Race/Ethnicity detail should probably be acknowledged before being ignored (caveat: this is a pet peeve). 

The results, conclusions, and Figure 1 should also focus on Hispanics, as they are mentioned in the text. There should be a paragraph in results about the Hispanics that mirrors that for Blacks, at least a sentence in the conclusions, and probably a 1C section for Hispanic data that mirrors 1B. My assumption is that the data showed no significantly different odds ratios for Hispanics, so all of this was stricken for not showing a relationship. If so, the data cannot reject lack of a health equity issue (null hypothesis). If one goes looking for a health equity issue for Hispanics and there isn't a visible relationship that supports one's assumptions for Hispanics, that should probably be explicitly stated or there is a risk of appearing to "cherry pick" to meet expectations coming into the scholarship. It might also be worth noting why other groups were not considered (e.g. Non-Hispanic Asians, which might flip the other direction). I'm assuming it was an issue of sample size, rather than an a priori assumption that only Blacks and Hispanics would be significant.

This dovetails into the bigger point regarding the Discussion. The implications of Figure 1B should guide more of the conclusions of the manuscript. I think the existing first two sentences at the start of the discussion are fine. But it should be spelled out that, when combining Black composition and urbanicity, only >42.5 was significant. The existing sentence points out there is a continuum for metropolitan areas, but it does not indicate that neither Suburban nor Rural were significantly different for Blacks. Nor does it note, apparently, that the authors could not find such a relationship for Hispanics. This should be noted, but the authors then have the opportunity to explain why a relationship was not found (brainstorming rather than statements of fact: data limitations, underreporting of Hispanics in Rural areas that use migrant workers, Blacks really are the special case here and being Hispanic/Latino doesn't matter, there's a difference between being Hispanic in some counties rather than others—kinds of work, illegal vs. citizen, etc.—that is obscuring a real relationship). The significant mortality odds ratio is almost self-explanatory (less people vaccinated means more die), but this is not explicitly stated. Nor is the meaning of the non-traditional vaccine distribution sites and availability of other vaccine distribution locations. Are non-traditional sites appearing in counties that otherwise would be 'vaccine deserts' and therefore indicate that decision makers realized they had a problem in these locations and shored up vaccine coverage? Does the less than 500m calculation matter at all in rural locations? All of this also feeds into your abstract conclusion, which could be more specific. 

Less pressing suggestions: 

Please include line numbers on the next draft. I can't give you specific feedback about typos, etc, because it is hard to direct you.

Please provide the reader with a stage setting of what kinds of vaccine sites are seen nationally and then dive into the data you have for those sites. For example, what about pop-up clinics, what about vaccines at long term care facilities? Indicate to me that you have considered/know about all of the options and how they fall into your categories.

PLOS Medicine may want outcomes first in the abstract and methods (I am guessing), but if not, it would be better to set the stage before going straight to them. 

Was it really defined as "indicator of Trump Win?"

The second paragraph of the results could be more explicit that it is discussing 'all 4 facility types', if that is what it is doing.

You might want to give more basic numbers (ideas rather than exact guidance follows): How many counties were you looking at? if there are some you had to remove, state that or throw up a map showing your data gaps to assure the reader that you are being close to comprehensive. Are you just looking at the US States? What about USVI, Puerto Rico, American Samoa, etc.? How many Blacks, how many Hispanics?

Other potential data issues that might be worth addressing: I can't think of exactly how it would affect your data overall, but people jump counties for vaccines (go into a city for work, the clinic is one county over, but close to home, etc.). Also, people jump channels (pharmacy to federal facility, etc.). Maybe this noise is self-limiting in your data, I don't know if this is worth addressing or not. If you have taken data from the beginning until now, do you need to account for individuals getting up to three doses at this point or successively wider age ranges for vaccination? Is there some kind of significant relationship for the J and J one shot vaccine going into a particular population or urbanicity that skews your results?

This is probably beyond your scope, but it would be interesting in a figure showing vaccines over time for Blacks and Hispanics. As COVID-19 has gone on, has the country done a better job at addressing inequities at the county level? Is there a way to display that?

Reviewer #3: 

Disparities in Distribution of COVID-19 vaccines across US Counties

Hernandez et al.

The authors present a study of COVID-19 vaccination sites, with the goal of evaluating fairness of vaccine access.

Major comments

Introduction

Please include a summary of or at least references to one of the many frameworks that were created prior to COVID-19 vaccine availability to support equitable distribution of vaccines, e.g., PMID: 3237895. In the discussion section, consider commenting upon our success at following the goals laid out in one or more of these frameworks. Additionally, it's important to recall that the US allowed COVID-19 vaccine eligibility in phases; this should be mentioned in the introduction as it likely affected placement of vaccine sites. 

The authors state, "to our knowledge, no nationwide studies have measured health equity in the actual distribution of COVID-19 vaccines to healthcare facilities;" however, studies taking a different approach to evaluation of health equity in vaccine distribution have been published and are easily accessible, e.g., PMID: 34213561. The authors should consider a more thorough literature review to provide the reader with a succinct summary of the work that has already been done on health equity and vaccine distribution in the US.

Methods and Results

As this study only evaluated vaccination sites in place as of May 2021, it's important to consider how eligibility for and availability of vaccines changed over time (https://www.cdc.gov/vaccines/imz-managers/downloads/Covid-19-Vaccination-Program-Interim_Playbook.pdf, see page 12), and to note that this varied by county. If possible, it would be helpful for the authors to compare their data to similar data later in 2021 or in early 2022. If this is not possible, the study should be clearly characterized as an early evaluation of vaccine sites, e.g., in the first sentence of the Results section should read, "…provided COVID-19 vaccinations as of May 2021." 

Consider showing the outcome(s)/independent variables stratified by the population types that were eligible for vaccination during their study period, which ended in May 2021 (therefore, only 3-4 months after COVID-19 vaccines were available to only certain portions of the population [e.g., healthcare workers who likely were vaccinated at their places of employment, and the elderly, who were likely vaccinated at their physician's office). In other words, during the time when vaccines were only available for healthcare workers and people >65, were vaccines easily accessible to the corresponding elderly portion of the population? 

Additionally, it would be helpful for the authors to comment upon the dates of their independent variable data. E.g., did they obtain the COVID-19 mortality data from the Johns Hopkins dashboard in or before May 2021 so that it temporally corresponded with their vaccine site data?

The table suggests that proportions of vaccine sites available to counties with varying levels of Black/Hispanic population did not vary significantly (e.g., for counties with black population <12.5%, 12.5%-42.2%, and >42.2%, the percent of healthcare facilities that served as vaccine sites was 60.5%, 63.9%, and 55.5%, respectively. It would be helpful for the authors to include a statistical comparison of the differences in these proportions and comment upon any significant findings, rather than merely mentioning the proportions of select groups in the Results section.

Discussion

Again, the authors' results should be characterized as "early" if they are unable to show later analyses for comparison. E.g., in the first sentence of the Discussion, they should write "...disparities in the early distribution…" 

Consider offering suggestions as to why the findings may be true. E.g., what barriers challenged availability of COVID-19 vaccination centers in the "hardest hit" communities? Were early COVID-19 vaccination sites targeted at locations with concentrated populations of people >65yo regardless of race because only people >65yo were eligible for vaccination at the time of the study?

It doesn't make sense to evaluate availability/accessibility of vaccine sites when most of the county population was not eligible for vaccination. For most of the study period, which ended in May 2021, only a relatively small proportion of the population was eligible in most counties (e.g., healthcare workers, the elderly, people with certain medical conditions). I think it's important that the authors discuss (and hopefully address by updating their data/analysis!) these limitations in the discussion section. In the last paragraph of the Discussion they provide a call to action that may not be necessary if vaccine equity did in fact improve after the study period. Ideally the authors would apply the same methods to evaluate health equity at vaccine sites at a later period as well, e.g., in late 2021 and/or early 2022.

Reviewer #4: OVERALL COMMENTS: 

This is an important and well-written study that examined the equity in vaccine districubtion across health care facilities and pharmacies in the US. The study examined equity in the distribution of COVID-19 vaccines across healthcare facilities and pharmacies in the US. They found the distribution of vaccines differed by geographic and racial composition. 

INTRODUCTION:

The introduction is too brief and does not fully expand on the scope and significance of the problem. Even a descriptive report on who received COVID-19 vaccines to date would be helpful. There had been a recent study on public opinion of vaccine distribution prioritization in 2021 that may be useful to cite. 

Would also be helpful to state why measuring health care facilities is important beyond who received the vaccines. You state this later in the manuscript, but this is important to describe in the introduction. 

In addition, the paper is focused on the health facilities that were registered to distribute. It may also be helpful to briefly mention the barriers to the health care facilities getting registered and how this may exacerbate disparities? 

METHODS:

Under data sources: 

Please operationalize vaccine distribution equity, this will help ground the results. 

Please define the acronyms for HOD and RHC. 

The authors indicate the study was exempt, was this determined by and IRB? If so, please state that.

Under independent variables:

It is not clear why election results were included. A sentence or two to explain this was included.

Can you provide an example of an alternative vaccine distribution location? The authors define what they were not (HOD, RHC and FQHCS), but not what they were. 

RESULTS:

In the figure, a key for the shapes (triangle, circle, square) would be helpful. 

DISCUSSION:

In the first sentence of the Discussion section. I believe it should read "…quantify disparities in the distribution locations of COVID-19 vaccines, …". 

The interpretation of the interaction terms should be mentioned. 

The data is limited to the first quarter of vaccine distribution to the general public (May 10, 2021). This should be discussed in the limitations. Distribution may have improved over time. 

Also worth mentioning the role of preference for where to access, resources in rural and safety-net locations. A location to distribute and resources to distribute are both important. 

The odds ratios are helpful but not sufficient. It is not clear what the impact is. Ground the results in how many lives are impacted by the odds. 

Reviewer #5: Overall, this article poses an interesting research question but lacks details and justification about their Methods which would be used to judge their Results. Specific comments include:

The Introduction is very sparse on why the variables they chose to investigate are important and relevant. This should be built out more.

This sentence need a citation: "County-level measures of racial/ethnic composition included proportion Black and proportion Hispanic, and were categorized in three levels following CDC methodology:(6) 1) below US average; 2) between US average and 95th county percentile; 3) above 95th county percentile."

Authors do not include their variable selection methodology for the final logistic regression model.

Authors do not include counts of facilities, etc. in the Results section write-up. This is important for readers to understand how big these sample sizes are and to determine if the Methods are sound.

There are no p-values in the text - unless excluding them is part of the journal style, this is essential to include.

What proportion of the U.S population falls into metropolitan vs. suburban vs. rural counties? This is an important comparator.

Reviewer #6: General impression: This is a brief report of findings on variability in distribution of vaccinates from eligible to distribution centers/pharmacies. The written presentation is clear and presents the data well. The data presentation is difficult to interpret without the scale (how many pharmacies and facilities were examined and considered eligible). The ability to post the publicly available data on the number of eligible pharmacies and their distribution would strengthen the ability to replicate the findings. 

Abstract: The abstract is written clearly and coveys the key findings well. One point that is not clear is the interpretation of the significant interaction between urbanicity and Black area-level composition - the interpretation of that interaction term should be described more clearly. 

Objectives: The study objectives are to understand the "likelihood of an eligible healthcare facility administering COVID-19 vaccines varied with the county-level racial composition and degree of urbanicity." However, the introduction reads that the study motivation is to better understand the "actual distribution of COVID-19 vaccines to healthcare facilities." These are related but different concepts with different policy implications. What led to the distribution of vaccines (structural problem)? Why did specific institutions or facilities administer vaccines (institutional or structural issue)? The objectives in the introduction might be helped by additional information on the hypothesis, and any evidence supporting the interpretation of the hypothesis.

Methods:

The methods do not describe how vaccination sites that were not listed on VaccineFinder were found. If these clinics were not included, this should be noted as a data limitation in the discussion section. The language for some of the concepts should be reviewed and clarified for an international audience (what is a Trump win? Is that to indicate Republican, Democratic or Independent candidate for elected office/president?) The limitation section should discuss some of the assumptions made calculating non-pharmacy locations (for example, Gillette Stadium, a large football field, was used to distribute vaccines in Massachusetts). 

Results: Throughout the results presentation, the Ns for the number of pharmacies/vaccine sites should be presented in addition to the percentages. Without a sense of the number of eligible pharmacies/vaccine sites, the data are difficult to interpret. The data presentation would be strengthened by the ability to verify the vaccine distribution sites (a link to these publicly available data).

Discussion: The limitations of the data should be described in greater detail as indicated in the methods section of this review.

[LINK]

---

## [Decision Letter · Decision Letter 2]

9 May 2022

Dear Dr. Hernandez,

Thank you very much for submitting your manuscript "Disparities in Distribution of COVID-19 Vaccines across U.S. Counties: A Geographic Information System-Based Cross-sectional Study" (PMEDICINE-D-21-04494R2) for consideration at PLOS Medicine. 

[LINK]

In light of these reviews, I am afraid that we will not be able to accept the manuscript for publication in the journal in its current form, but we would like to consider a revised version that addresses the reviewers' and editors' comments. Please pay particular attention to reviewer #3's second concern regarding comparison of data at multiple pandemic time points and extracting the required data for later pandemic time points from the same sources you have used for the primary analysis. Obviously we cannot make any decision about publication until we have seen the revised manuscript and your response, and we plan to seek re-review by one or more of the reviewers. 

We expect to receive your revised manuscript by May 30 2022 11:59PM. Please email us (plosmedicine@plos.org) if you have any questions or concerns.

We look forward to receiving your revised manuscript. 

Sincerely,

Beryne Odeny, 

PLOS Medicine

plosmedicine.org

Comments from the reviewers:

Reviewer #1: The authors have responded to each comment in turn, and have suitably included social vulnerability index into the analysis.

Reviewer #2: Thanks to the authors for addressing my comments and those of others. Some of them may have seemed exasperating, but I think this version is clearer for the adjustments.

Line Edits:

Line 81, Should White be non-Hispanic White? "among racial/ethnic minority groups than White individuals" This occurs again later. It's fine to use short-hand after first use, particularly if you spell that out, like you do for "Blacks" being "non-Hispanic Blacks," but if there is ambiguity, I'd at least define it the first time. You could check all your race references for this. For instance, I might explain blacks again when first mentioned in the discussion. Line 218 could use some precision for White and Black too. Or change throughout so that "Black" and "White" never stand alone (and consider defining what Hispanic means?).

Line 105-106.When you are talking about LTCFs and how they aren't included, you could talk about the other types (like stadiums) that are not included? Perhaps move up Line 159 -165.

Line 113. Add the month and year that adults got access to the vaccine.

Line 156-157 Rather than fixating on one of the parties, you could say you created a variable that said whether Trump or Biden won the election, unless there was someplace another candidate one. This would keep the audience from reading between the lines and getting tangled up in politics of COVID-19 and health equity.

Line 172 I try to avoid sentences that start with dependent "because" clauses.

Line 212 comma after metropolitan (trying to leave the line-editing to the line editor, sorry). Line 214 needs a comma after rural too.

Line 245 "Non-Hispanic Black and Hispanic" These could use a noun, like "people" or "minority groups."

Line 248 identification identify 

Line 251 delivered "at" each site. Same issue for line 253, up to you.

Line 255 suggested adjustment: "so it was impossible to define whether institutional or structural reasons led some facilities to not become COVID-19 administration sites". Sentences that lead with negatives always sound a bit awkward, but I don't how to fully remove it here.

Line 258 level","

Line 259-262. This sentence could use a rewrite. Too many clauses, too many words.

Line 263 The discussion "regarding the causes of" lower uptake. I hope the discussion isn't focused on mistrust. Makes peer review more difficult, if so. 

Line 265 starts with almost a pun, I like it.

Line 266 Does this journal allow references in the middle of sentences? if so, never mind.

Reviewer #3: The authors responded adequately to many of the reviewers' comments, and specifically to many of my own previous comments. However, two remaining glaring concerns are as follows:

1. Is it correct to draw conclusions about (and quantify) vaccine equity based only on number of sites eligible vs number of sites that actually distributed vaccines without the context of how many vaccines were distributed? Eg, even if 10 sites in a given area are eligible to distribute vaccines but only 5 of the sites actually distribute the vaccines, as long as those 5 sites distribute enough vaccines to the area's population then equity is achieved despite the fact that all 10 eligible sites did not distribute vaccines. As the authors used county-level measures for race/urbanicity, etc, why not additionally evaluate county level measures for number of COVID vaccines distributed during the study period? These county level data certainly exist, though they may exist outside the vaccine finder dataset. 

2. The authors have made clear that they are unable to compare data at multiple pandemic time points, though the reason why they are unable to extract data for later pandemic time points is unclear (it should be available from the same sources the authors used to evaluate the May 2020 time point, eg from CDC's vaccine finder). Longitudinal evaluation of the data that the authors have put forward is essential to determine whether geographic disparity in vaccine distribution existed during the pandemic and whether or not it improved. Given the number of unknown variables that may have influenced the authors' finding that all eligible vaccine distribution sites did not distribute vaccines early in the pandemic, the utility of this stand-alone/cross sectional finding is unclear. If this finding persisted over time--particularly in the context of comparatively fewer vaccines being distributed to underserved populations living in a given county over time--a more meaningful/impactful conclusion could be drawn.

[LINK]

---

## [Editor Report · Decision Letter 3]

23 Jun 2022

Dear Dr. Hernandez,

Thank you very much for re-submitting your manuscript "Disparities in Distribution of COVID-19 Vaccines across U.S. Counties: A Geographic Information System-Based Cross-sectional Study" (PMEDICINE-D-21-04494R3) for review by PLOS Medicine.

I have discussed the paper with my colleagues and the academic editor. I am pleased to say that provided the remaining editorial and production issues are dealt with we are planning to accept the paper for publication in the journal.

[LINK]

We look forward to receiving the revised manuscript by Jun 30 2022 11:59PM.   

Sincerely,

Beryne Odeny, 

PLOS Medicine

plosmedicine.org

Requests from Editors:

1) Please temper claims of primacy in the discussion (i.e., “… because it is the first to analyze equity in the distribution…”) by stating, "to our knowledge" or something similar.

Notes from Academic Editor:

I think the reviewers' remaining (reviewer #3) concerns are justified and I also think the authors’ responses are reasonable. Nevertheless, I do think the authors should acknowledge some of these issues in their limitations. 

On my read, the reviewer is questioning in part how meaningful the fraction of eligible facility delivering vaccines is, because this does not account for the volume of vaccines given per site, and a geography could be “covered” even if fewer facilities offered if those that did offered enough and were accessible enough. The authors respond that they are interested in the “distribution of vaccines” — I find this reasonable but not entirely convincing for two reasons. First, they say that whether a facility distributes at all is a more meaningful indication of equity in the supply, but it is not clear to me that regional demand does not influence whether a facility distributes at all. In other words, do we know that distribution as has been measured here is not influenced by demand? I do, nevertheless agree, that the “distributed or not” outcome is still valuable.

Second, I do think that the authors need to be clear that they are looking at a particular moment in time. The equity of distribution of COVID related services (whether testing or vaccines) changed dramatically over time. Regardless of how it ended up, the initial geographical distribution of vaccine supply is important, but it should be noted if this was during a particular swath of time. May 10, 2021 was very early in the vaccine roll out. In fact the whole paper needs to be clear that this is disparities in the initial roll out. 

[LINK]

---

## [Editor Report · Decision Letter 4]

6 Jul 2022

Dear Dr Hernandez, 

On behalf of my colleagues and the Academic Editor, Dr. Elvin Hsing Geng, I am pleased to inform you that we have agreed to publish your manuscript "Disparities in Distribution of COVID-19 Vaccines across U.S. Counties: A Geographic Information System-Based Cross-sectional Study" (PMEDICINE-D-21-04494R4) in PLOS Medicine.

PRESS

Sincerely, 

Beryne Odeny 

PLOS Medicine